# High Incidence of Intracerebral Hemorrhaging Associated with the Application of Low-Intensity Focused Ultrasound Following Acute Cerebrovascular Injury by Intracortical Injection

**DOI:** 10.3390/pharmaceutics14102120

**Published:** 2022-10-06

**Authors:** Evgenii Kim, Jared Van Reet, Hyun-Chul Kim, Kavin Kowsari, Seung-Schik Yoo

**Affiliations:** 1Department of Radiology, Brigham and Women’s Hospital, Harvard Medical School, Boston, MA 02115, USA or; 2Department of Artificial Intelligence, Kyungpook National University, Daegu 37224, Korea; 3Department of Mechanical Engineering, Massachusetts Institute of Technology, Cambridge, MA 02139, USA

**Keywords:** brain, ultrasound, drug delivery, interstitial tracers, hemorrhaging, glymphatic

## Abstract

Low-intensity transcranial focused ultrasound (FUS) has gained momentum as a non-/minimally-invasive modality that facilitates the delivery of various pharmaceutical agents to the brain. With the additional ability to modulate regional brain tissue excitability, FUS is anticipated to confer potential neurotherapeutic applications whereby a deeper insight of its safety is warranted. We investigated the effects of FUS applied to the rat brain (Sprague-Dawley) shortly after an intracortical injection of fluorescent interstitial solutes, a widely used convection-enhanced delivery technique that directly (i.e., bypassing the blood–brain-barrier (BBB)) introduces drugs or interstitial tracers to the brain parenchyma. Texas Red ovalbumin (OA) and fluorescein isothiocyanate-dextran (FITC-d) were used as the interstitial tracers. Rats that did not receive sonication showed an expected interstitial distribution of OA and FITC-d around the injection site, with a wider volume distribution of OA (21.8 ± 4.0 µL) compared to that of FITC-d (7.8 ± 2.7 µL). Remarkably, nearly half of the rats exposed to the FUS developed intracerebral hemorrhaging (ICH), with a significantly higher volume of bleeding compared to a minor red blood cell extravasation from the animals that were not exposed to sonication. This finding suggests that the local cerebrovascular injury inflicted by the micro-injection was further exacerbated by the application of sonication, particularly during the acute stage of injury. Smaller tracer volume distributions and weaker fluorescent intensities, compared to the unsonicated animals, were observed for the sonicated rats that did not manifest hemorrhaging, which may indicate an enhanced degree of clearance of the injected tracers. Our results call for careful safety precautions when ultrasound sonication is desired among groups under elevated risks associated with a weakened or damaged vascular integrity.

## 1. Introduction

The blood–brain barrier (BBB), formed by tight junctions between cerebral vascular endothelial cells, along with glia limitans, prevents cells and large molecules from entering the brain parenchyma and plays a defensive role against various toxins and microorganisms [1,2]. However, the BBB also significantly limits the delivery of therapeutic agents into the brain, only allowing small (<500 Da) molecules to cross [3]. To directly introduce pharmaceutical agents into the brain, a needle or catheter is placed into the brain parenchyma or over the cortical surface [4,5]. This convection-enhanced delivery (CED) technique, due to its invasive nature, is mainly performed to deliver chemotherapeutic agents [6,7], oncolytic viruses [8,9], and monoclonal antibodies [10,11] for the treatment of brain tumors, whereby the drug distribution is managed by the flow rate [12] and catheter shape [13]. The drug distribution can be further propelled by the acoustic streaming effects induced by the application of ultrasound pressure waves to the area of the drug entry [14,15].

Ultrasound parameters that enhance CED in non-human primates (NHP) include a 140 min-long continuous application of ultrasound (236 kHz and a 3 kPa peak-to-peak pressure), which has been shown to increase the distribution of a gadolinium contrast agent and Evans blue tracer in the subcortical regions [16], and a 60 min application of pulsed ultrasound (1 MHz, 20 μs pulse duration, 2.5% duty cycle and a 1.24 MPa peak-to-peak pressure) that was shown to enhance the distribution of an intracerebrally-injected gadolinium-based contrast agent [17]. Although these studies did not show signs of brain tissue damage associated with the use of ultrasound, the safety of an ultrasound application in CED techniques, especially following the disturbance of the cerebrovascular system caused by direct cortical injection, remains largely unexplored, necessitating its assessment.

In the present study, we examined the effect of an FUS that was given a short time (10 min) after the intracortical injection of interstitial fluorescence tracers in rats. The procedure of an intracortical injection, frequently adopted in the CED modalities, typically does not lead to any significant intracerebral hemorrhaging (ICH), whereby the rate of the insertion/removal of the stereotactic injection needle (typically 30-gauge (30G) or thinner) is controlled to allow the brain tissue to close following needle retraction; however, the procedure would leave a degree of localized insults to the brain parenchyma and surrounding vasculature [18,19]. The intracortical injection technique is also widely adopted to investigate the brain lymphatic clearance of interstitial cerebrospinal fluid solutes [20]. As the FUS parameters used in the study neither elevate the risks of tissue damage nor disrupt the BBB in the absence of intracortical injection [21], we primarily examined the degree/presence of macroscopic hemorrhage in the brain tissue. In addition, the volume of the tracer distribution was evaluated to probe the effects of the FUS on the interstitial solute transport.

## 2. Materials and Methods

The study was approved and conducted according to the rules set forth by the Institutional Animal Care and Use Committee (IACUC). A total of 15 Sprague-Dawley (SD) rats (all male, eight weeks-old; 289.0 ± 14.6 g) were randomly divided into two groups, one that received FUS (*n* = 10, 290.0 ± 13.3 g and noted as ‘FUS+’) and the other that did not (*n* = 6, 286.7 ± 17.4 g and noted as ‘FUS−’). All animals were anesthetized using an intraperitoneal injection of ketamine/xylazine (80:10 mg/kg dose). After removing the fur on the head using a clipper followed by an application of depilation lotion, lidocaine HCl (5 mg/kg, diluted to 0.5% with sterile water) was subcutaneously injected beneath the scalp to reduce any pain associated with the surgical procedure for the intracortical injection. The animal was placed on a stereotactic frame (ASI Instruments, Warren, MI, USA), an incision was made in the midline over the skull and the bregma was identified.

The tracers used were a medium-molecular weight (M_W_) Texas Red ovalbumin (OA, 45 kDa) and a high-M_W_ fluorescein isothiocyanate (FITC)-dextran (FITC-d, 2000 kDa), constituted at a concentration of 0.5 wt% in artificial CSF (aCSF, Tocris Bioscience, Bristol, UK). A burr hole (~1 mm diameter) was carefully drilled over the skull without penetrating the dura, at 2 mm caudal and 3 mm right of the bregma. A 30G needle attached to a 10 μL gastight syringe (Hamilton, Reno, NV, USA) was slowly inserted through the burr hole (at a rate of 1 mm/30 s), reaching the target depth of 3 mm (at the interface of the cortical area and corpus callosum, Figure 1a). A 30G needle is widely used for rodent interstitial injection procedures [22,23,24]. The tracer was then delivered in a range of volumes (0.5, 1 and 2 µL; *n* = 2, 7, and 6, respectively) for 2–5 min using a microinjection syringe pump (Legato 100, KD Scientific, Holliston, MA, USA). After the injection, the needle remained in place for an additional 10 min and was then slowly retracted over a period of 6 min to allow the tissue to close shut (schematics shown in Figure 1b). The procedure was adapted from previous publications on interstitial injections in rodent brains [20,25,26,27], whereby the closure of the brain tissue was verified by the absence of a backflow of the tracer solution from the injection site retraction. Once the needle was removed, the burr hole was filled with bone wax (Ethicon Inc., Raritan, NJ, USA). The wound was not sutured given the non-survival nature of the experiment, and the animal was moved to another robotic stage where the FUS was guided stereotactically. The time between the needle withdraw and the start of the sonication was maintained at 10 min across all animals.

The FUS was transcranially delivered to the injection site from the top of the head through a polyvinyl alcohol (PVA) hydrogel acoustic coupler. The acoustic pressure field generated by the 200 kHz transducer (Ultran Group, State College, PA, USA) was directly mapped in a degassed water tank using a needle hydrophone (HNC-200, Onda Corp, Sunnyvale, CA, USA) mounted to the 3-axis robotic stage (Bi-Slides, Velmex Inc., Bloomfield, NY, USA) at 1 mm step resolutions. The focus was formed 11 mm away from the exit plane of the transducer and was 5 mm in diameter and 15 mm in length (with the full width defined at 90% maximum of the pressure, Figure 1c). The use of a 200 kHz ultrasound frequency was based on its advantages over higher frequencies toward translative applications, whereby excellent transcranial transmissions of acoustic pressure waves (~40%) have been observed in humans [28,29,30]. A higher frequency, on the order of 600 kHz, may yield a tighter focus (given the shorter wavelength); however, it would suffer from a greater insertion loss during the transmission through thicker skulls such as those of humans. The location of the acoustic focus was aligned to the injection site using a 3-axis robotic sonication platform (MSL Series, Newmark Systems, Rancho Santa Margarita, CA, USA). The sonication was given with a 100 ms pulse duration every 1 s (10% duty cycle) for 30 min. As an excessive intensity of ultrasound may eventually impose undesired mechanical effects such as cavitation [31], we used a low-intensity FUS, given at a spatial-peak pulse average intensity (I_SPPA_) of 5 W/cm^2^ (a corresponding spatial-peak temporal average intensity (I_SPTA_) of 500 mW/cm^2^ and a mechanical index (MI) of 0.8, corresponding to a rarefactional pressure of 386 kPa). This intensity was far below the regulatory (i.e., the FDA) guideline parameters on clinical ultrasound imagers, including transcranial Doppler sonography devices [32]. The brain lymphatic clearance is known to be affected by cardiac pulsation, necessitating the measurement of the respiratory rate, heart rate, and SpO_2_ at the onset and at the end of the FUS sonication. As a control condition, six SD rats (all male, weighing 286.7 ± 17.4 g) underwent the same interstitial tracer injection procedure and were placed under the sonication platform, but without receiving any sonication.

All the animals were immediately euthanized using a transcardial perfusion with 4% paraformaldehyde (PFA) in phosphate buffered saline (PBS). More specifically, ~150 mL of normal saline was perfused with a 25 mL/min flow rate pump, then, the solution was exchanged to PFA at room temperature (~20 °C). The skull was extracted and underwent another 24 h immersion fixation in PFA before harvesting. The harvested brain was cut to encompass the injection site in a 4 mm-thick block, then embedded in an agar gel block (3 wt% in distilled water, A4018, Millipore Sigma, Waltham, MA, USA). The block was then cut in 200 µm-thick slices using a vibratome (PELCO easiSlicer, Ted Pella, Redding, CA, USA), and the sections were imaged using a wide field-of-view (1×) fluorescent microscope (TS100, Nikon, Tokyo, Japan) for visualization of the OA and FITC-d.

To estimate the volume of tracer distribution in the brain section, nine slices of fluorescent images that included the injection site in the middle (8-bit monochrome) were segmented using a threshold of 3 times the median absolute deviation value above the median (the ‘isoutlier’ function as implanted in MATLAB software, version 2022a, Mathworks, Natick, MA, USA).

The hemorrhage volume was estimated from the corresponding bright-field images. To do so, each image was first split into RGB (red–green–blue) tricolor channels, and the blue channel image was normalized with respect to the corresponding red channel image, pixel-by-pixel. As normal brain tissue has similar color spectral densities for the blue and red channels, whereas hemorrhage has a higher red spectral intensity per pixel (due to the presence of red blood cells (RBCs)), a threshold condition was then applied ≤0.5 to delineate the pixels indicating the presence of hemorrhage. The counted pixels in each image were converted to volume (529 pixels = 2 nL; calibrated using a reference hemocytometer image).

A statistical analysis was performed using the MATLAB software (Statistics and Machine Learning Toolbox). The animals’ weights, respiratory/heart rates and SpO_2_ between the two groups (i.e., FUS+ and FUS−) were compared using a *t*-test. The within-group difference in the respiratory/heart rates/SpO_2_ as well as in the tracer volume distribution was assessed using a paired *t*-test. The presence of dependency between the injected tracer volume and the occurrence of hemorrhaging was evaluated by a Chi-square test. A nonparametric Kruskal–Wallis test followed by a Fisher’s least significant difference (LSD) post hoc analysis was used to compare the hemorrhage size among the groups. The difference in the tracer volume between the FUS+ and FUS− conditions was examined using a Mann–Whitney U test. The statistical significance was set at *p* < 0.05.

## 3. Results

The two groups of rats were indifferent in terms of weight (two-tailed *t*-test, *p* = 0.68). Examples of the bright-field and fluorescent images from a control, unsonicated rat (receiving 1 μL of the tracer) are shown in Figure 2a (top row). In the control rats (FUS− group), the OA and FITC-d were distributed around the injection site, with a wider distribution of the OA (21.8 ± 4.0 µL) compared to that of the FITC-d (7.8 ± 2.7 µL; paired two-tailed *t*-test, and *p* = 0.00002), all without the presence of any hemorrhaging (animal-specific bright-field sectional images at the injection site are shown in Appendix A).

We found that four out of the nine sonicated rats (44.4%) showed hemorrhaging (example shown in Figure 2b middle row; noted as FUS+/Hem+). The occurrence of hemorrhaging in the FUS group was not affected by the injected tracer volume (a Chi-square test, d.f. = 8, and *p* = 0.78). We note that the hemorrhaging was detected not only at the site of the tracer injection, but also far away from the site, extending in the rostral and caudal directions and to the hemisphere contralateral to the injection (Figure 2b, white arrows). In these animals, the distribution of the hemorrhage and the interstitial tracer did not coincide.

A paired comparison of the respiratory/heart rates, and SpO_2_ measured at the onset and completion of the FUS within each group showed no differences (Table 1, paired two-tailed *t*-test, and all *p* > 0.1). These physiological variables, when compared between the groups (i.e., the FUS and control), were also indifferent (two-tailed *t*-test, and all *p* > 0.17), indicating that a relatively short monitoring period (~30 min) before sacrifice did not ramify into detectable changes in the observed physiological variables, despite the presence of massive hemorrhaging among the sonicated rats.

The volume of tissue that indicated the presence of hemorrhaging from the sonicated group (FUS+/Hem+, *n* = 4) was 29.5 ± 21.0 µL, whereas only a miniscule amount of bleeding was detected from the remaining sonicated rats (FUS+/Hem−, 0.01 ± 0.006 µL; *n* = 5) and the control group (FUS− group, 0.022 ± 0.017 µL, *n* = 6; Figure 3a). The Kruskal–Wallis analysis showed a significant difference in the hemorrhage size among the experimental groups (H(2) = 9.42, *p* < 0.01). The post hoc analysis using the Fisher’s least significant difference (LSD) further revealed a difference between the FUS+/Hem+ group and the non-hemorrhage groups (*p* < 0.01). No statistical difference in the hemorrhaging volume was found between the FUS+/Hem− and the FUS− group (*p* = 0.28).

Since hemorrhaging obscured the interpretation of the OA and FITC-d fluorescence, the tracer distribution between the FUS and control conditions was compared only among the animals that did not show hemorrhaging (*n* = 5; an example is shown in Figure 2c). The comparison revealed that sonication reduced the volume of the tracer distribution for both the OA (12.9 ± 7.4 µL) and FITC-d (3.5 ± 4.0 µL; one-tailed Mann–Whitney U test, and both *p* < 0.05; Figure 3b) compared to the control rats that did not receive the FUS (19.7 ± 3.9 µL OA and 8.5 ± 2.8 µL FITC-d).

## 4. Discussion

The therapeutic utilities of ultrasound for promoting healing in musculoskeletal conditions, such as in tendon repair or bone healing, are well-documented [33,34]. Over the last decade, the use of FUS for neurotherapeutic applications has been rapidly expanding. Other than functional neurosurgical approaches to ablate the brain tissue using the application of high intensity focused ultrasound (HIFU) [35,36], the use of low-intensity FUS has been demonstrated across various arenas, for example, the enhancement of CED effects, the release of drugs through ultrasound-sensitive drug-loaded carriers, and the microbubble (MB)-mediated disruption of the BBB for the delivery of large-M_W_ macromolecules and biological agents [14,37,38]. Combined with the application of the MBs, FUS is known to enhance the effect of thrombolytic agents (e.g., tissue plasminogen activator, tPA) [39,40]. The mechanical waves from low intensity ultrasound have also been explored to unbind neurological drugs (such as phenytoin, an anti-epileptic agent) from plasma proteins to increase their local bioavailability [41]. Neurostimulation is another promising area of low-intensity FUS, with the ability to modulate the excitability of highly localized neuronal tissue deep inside the brain [42,43]. Undoubtedly, FUS provides an unprecedented opportunity for multi-faceted neurotherapeutics [44].

The use of mechanical pressure waves produced by FUS initially incited concerns over a potential heat generation and cavitation-related tissue damage. Heat generation can be readily avoided by applying the sonication with sufficient time intervals, allowing the heat to dissipate; however, minor local hemorrhaging has been observed in several studies during the BBB opening [45,46], while the presence of hemosiderin and RBC extravasation have also been identified from brain stimulation using FUS [47,48]. Therefore, cavitation-related effects on neuronal tissues and vasculature demand further interrogation.

In the present study, high rates of ICH were observed when the FUS was locally delivered to rat brains for 30 min, a short time (10 min) after retracting the injection needle. In contrast, the rats that were not exposed to sonication, which otherwise underwent a same intracortical injection procedure, did not show any signs of hemorrhaging. This finding opposed our anticipation that sonication, given at a low intensity compatible with most ultrasound imaging procedures, would not affect the site of an acute injury resulting from the microinjection procedure.

As to the potential causes of the hemorrhaging, we speculate that acoustic pressure waves might have exacerbated the integrity of the micro- and macro-scopic cerebral vessels following the needle injection. We note that the sonication parameters and setup, which were identical to the ones used in our previous work in promoting the transport of an intracisternally-injected CSF tracer into the brain parenchyma in rats, neither altered the BBB integrity nor yielded any signs of brain tissue damage (including any hemorrhaging) when applied in the absence of intracortical injection [21]. A minor reverberation within the cranial cavity is expected; however, the use of a 386 kPa peak rarefactional pressure (Pr) would not elicit inertial cavitation. Thus, we believe that the hemorrhaging observed in the present study was not likely to be associated with a standing wave formation or inertial cavitation.

Regarding the observation that hemorrhaging was detected in the brain hemisphere contralateral to the sonication, we conjecture that the compromised vasculature by the needle insertion was not sealed against the hemodynamic pressure due to the application of local acoustic pressure waves. The leaky vasculature, thus, might have resulted in hemorrhaging into the hemisphere opposite to the injection site. We cannot, however, completely rule out the contribution from the PFA perfusion which may have propelled extravasated blood cells further to reach far away from the wound site. The snap-freeze fixation technique [49] can be considered as an alternative to visualize the distribution of a hemorrhage.

It is important to note that ICH has not been observed in rodent CED studies in which ultrasound was administered simultaneously with an intracortical drug infusion through catheters that remained in place (without retraction) [14,15]. In humans, although ultrasound has not been used in conjunction, a catheter having a much greater outer diameter (e.g., a 2.1 mm outer diameter) than those used in these animal studies has been utilized for intraparenchymal drug delivery in clinical applications without causing ICH [50,51]. We surmise that the remaining catheters in these studies might have sealed/plugged the leaky vasculature. Additional time to allow for tissue healing around an inserted catheter may further reduce the likelihood of hemorrhaging. The non-sonicated rats from the present study did not show any hemorrhaging, which indicates that their vasculature was sealed against cardiovascular pressure in ~40 min following the needle retraction; therefore, it is reasonable to postulate that FUS, applied following a sufficient time period after a given local injury would not cause hemorrhaging. We acknowledge that the healing mechanism following injury to the brain from an intracortical needle injection would differ between rats and humans. For example, the rodent brain is primarily healed through wound contraction rather than re-epithelialization, which has been identified as a main healing mechanism in humans [52]. Moreover, the rodent brain lacks sulci and gyri, where the extensive existence of perivascular space (PVS) is identified in gyrencephalic brains [53]. The vascular closure time will be dependent on many factors, such as the extent of injury (e.g., imposed by various needle sizes), sonication parameters, and animal species; thus, this demands further examination.

The spatial distribution of the injected tracers in the unsonicated rats showed patterns consistent with previous investigations, including a more extensively distributed OA (45 kDa) than FITC-d (2000 kDa), which agreed well with the size-dependent brain lymphatic transport of the interstitial solutes [54]. However, among the rats that showed hemorrhaging, the distribution of the hemorrhage and interstitial tracer did not coincide (Figure 2b). This finding indicates that large-sized RBCs in the blood traveled further into different brain regions compared to the injected tracers having much lower M_W_ values. This observation may suggest the existence of different routes for extravasated blood and interstitial fluid. For example, once tracers enter the brain parenchyma, they are separated from the vascular system and move through the ‘brain lymphatic clearance pathway’, i.e., the perivascular space (PVS), being mediated by aquaporin 4 (AQP4) channels [55,56,57]. PFA transcardial perfusion, which would primarily affect the flow within the vascular system, can propel the extravasated RBCs to different parts of the brain, yielding the ICH pattern observed in the present study.

We also found that a decreased tracer volume was associated with a weak tracer fluorescence among the non-hemorrhagic rats that received sonication (Figure 2c). Based on recent evidence demonstrating the effects of FUS in promoting CSF/interstitial solute transport [21,58], along with a close anatomical and transport connectivity between the brain parenchyma and the PVS [53], we postulate that the local presence of acoustic streaming created by the sonication might have facilitated the tracer transport by promoting a bulk flow along the PVS, possibly ramifying into their clearance or dilution of fluorescence. Since the image analysis technique used in the present study (i.e., vibratome sectioning) severely limited the examination of the tracer distribution beyond a few millimeters from the injection site, assessments of the tracer distribution across the entire brain volume, including the tracers drained into the cervical lymph nodes, are warranted to validate this hypothesis. For example, the dynamic contrast-enhanced magnetic resonance imaging (MRI) that accompanies the use of MR-sensitive interstitial tracers may help to capture the spatiotemporal features of the tracer transport by the application of FUS.

A limitation of this work is that the PFA perfusion, used to fixate the brain tissue, may have shrunk the PVS [59], reducing the fluidic movement along the PVS, which in turn potentially confounded the movement of both the interstitial tracers and extravasated blood cells. An immersion fixation without the perfusion [60] can be considered as an alternative to reduce the confounders associated with perfusion fixation; however, this would be met with unknown contributions from uneven rates of fixation across the brain volume (including the PVS) [61]. Since there is a limited choice of tissue fixation methods, the in vivo monitoring of vascular leakage without the need for tissue fixation (such as two-photon microscopy [62]) or dynamic MRI using a T_2_* sequence (sensitive to hemoglobin [63]), would enable monitoring of the actual spatiotemporal features of ICH processes. We also acknowledge that there are techniques other than intracortical injection to induce an acute cerebrovascular injury in rodents. The middle cerebral artery occlusion or photo thrombosis technique is an alternative approach to provide acute cerebrovascular damage mimicking ischemic stroke conditions [64,65]. Further investigation on the safety of FUS following these alternative models can be considered in future work. Despite the low likelihood of inertial cavitation under the present experimental conditions, passive cavitation detection (PCD) during sonication [66,67] would help reduce the associated safety risks.

The overarching finding of our study is that the sonication parameters that have been deemed safe (i.e., being low-intensity/pressure, and nonthermal FUS), in the absence of any microbubbles (MBs), may cause unexpected hemorrhaging in the brain following intracortical injection (which is commonly used to introduce interstitial tracers to the brain without hemorrhaging). As the use of ultrasound in the therapeutic space is rapidly increasing, the present study raises important safety concerns for the clinical use of FUS, not only for CED applications but also for FUS-based brain stimulation techniques in neurorehabilitation after brain injury [68]. Since stroke- or tumor-related brain damage may compromise the mechanical integrity of the macro- and micro-scopic tissue/vascular environment (e.g., brain edema or necrotic/liquefaction changes) [69,70,71,72], FUS may significantly increase the risk of ICH. Age-dependent, unknown risk factors of ultrasonic brain stimulation may also exist in elderly adults [73,74,75], including patients with cerebral implants (such as brain shunts or aneurysm clips) who may receive unavoidable neurovascular insults. Considering the changes in brain tissue properties associated with these patient groups, our findings suggest the need for a careful safety assessment prior to intervention, and further investigation is required on evaluating the effects of FUS during the time course of recovery from brain injuries or pathological changes.

## Figures and Tables

**Figure 1 pharmaceutics-14-02120-f001:**
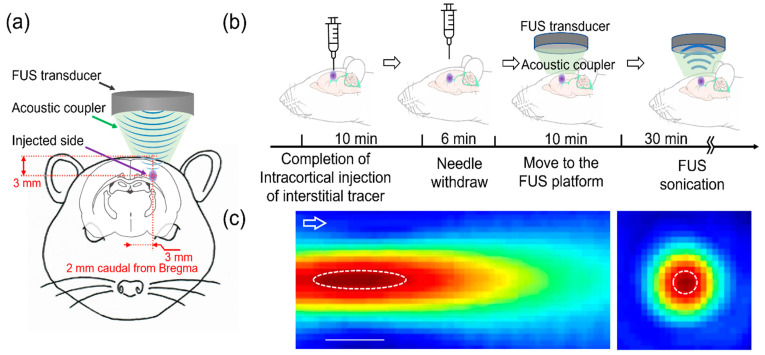
Schematics of the experimental procedure and the acoustic pressure map. (**a**) Descriptive drawing of the tracer injection site and sonication location. (**b**) Fluorescent tracers in artificial cerebrospinal fluid (aCSF) were intracortically injected and a focused ultrasound (FUS) was transcranially applied to the injected site. (**c**) Pseudo-color pressure map along the longitudinal and transverse plane of the FUS focus. The pressure contours defined at full width at 90% maximum are depicted by the dotted line (bar = 10 mm). The arrow indicates the direction of sonication.

**Figure 2 pharmaceutics-14-02120-f002:**
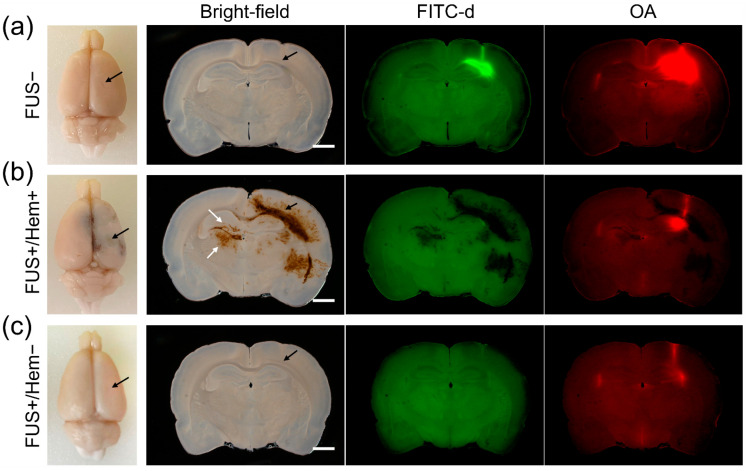
Brain imaging from different experimental groups. (Left to right), exemplar macroscopic images of extracted brains before sectioning, bright-field, ovalbumin (OA) and fluorescein isothiocyanate dextran (FITC-d) images from (**a**) a control, unsonicated rat (FUS−), (**b**) a rat that received FUS showing the intracerebral hemorrhage (FUS+/Hem+), and (**c**) without any hemorrhaging (FUS+/Hem−). Black arrows indicate the needle injection site. The white arrows specify the hemorrhage detected in the hemisphere opposite to the injection. Scale bar = 2 mm.

**Figure 3 pharmaceutics-14-02120-f003:**
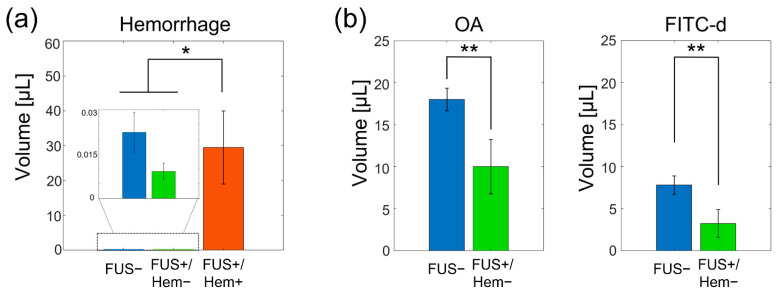
Comparisons of hemorrhage and tracer volume. (**a**) The hemorrhage volume obtained from the FUS− (*n* = 6, in blue), FUS+/Hem− (the rats without showing the hemorrhage; *n* = 5, in green), and FUS+/Hem+ (*n* = 4, in red) groups. (**b**) Tracer volume comparisons between the FUS− (in blue) and the FUS+/Hem− group (in green) for each tracer type. A statistically significant difference is indicated by ‘*’ (Fisher’s least significant difference (LSD) post hoc analysis followed by the Kruskal–Wallis test, *p* < 0.05), and ‘**’(one-tailed, Mann–Whitney U test, *p* < 0.05). The error bars indicate the standard error.

**Table 1 pharmaceutics-14-02120-t001:** Weight and vital signs from the animals. Respiratory/heart rates (RR and HR), and SpO_2_, measured at the onset and completion of FUS. Std: standard deviation.

		Weight (g)	RR (Breaths per Minute)		HR (Beats per Minute)		SpO_2_ (%)	
			Beginning	End	Beginning	End	Beginning	End
FUS+	Mean	290.0	55.6	55.1	206.1	196.4	85.2	87.3
	Std	13.3	8.1	6.4	23.3	14.5	4.8	5.9
FUS−	Mean	286.7	57.3	56.3	217.8	207.3	88.8	88.5
	Std	17.4	4.8	3.2	36.3	29.8	4.4	2.7

## Data Availability

The data presented in this study are available on request from the corresponding author. The data are not publicly available due to institutional restrictions on data sharing.

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
