# Peer review of "High Incidence of Intracerebral Hemorrhaging Associated with the Application of Low-Intensity Focused Ultrasound Following Acute Cerebrovascular Injury by Intracortical Injection"

_pharmaceutics, 2022, doi:10.3390/pharmaceutics14102120_

Round 1
Reviewer 1 Report (Previous Reviewer 2)
The authors mainly replied to the comments, the paper is suitable for publication
Author Response
Please see the attachment.

Reviewer 2 Report (New Reviewer)
In this manuscript, Kim et al. showed that FUS applied shortly after the intracortical injection caused ICH, which calls for safety precautions when using sonication with weakened or damaged vascular. However there are several questions must be addressed:
1. The authors should re-organize the abstract and introduction to:
(1) highlight what the novelty and significance of this research are. It is difficult to tell why studying the safety of intracortical injection and FUS is important without sufficient background.
(2) Sentences in the abstract such as “Texas Red ovalbumin (OA) and fluorescein isothiocyanate-dextran (FITC-d) constituted in artificial cerebrospinal fluid (aCSF) were used as interstitial tracers. Low-intensity FUS, delivered at an acoustic intensity compatible with ultrasound imaging, was applied to the injection site for 30 min in a pulsed fashion. The brain was then harvested through transcardial perfusion of paraformaldehyde,” and in the introduction such as “Low-intensity FUS was transcranially applied to the injection site shortly after (10 100 min) needle retraction to assess the effects of sonication” are very detailed experimental steps that should be moved to the method.
(3) The authors should clearly state the role of intracortical injection in this study. Is it to create cerebrovascular damage or for the CED? Because both appeared in the manuscript.
(4) Instead of saying what the authors initially hypothesized, it should be stated what is found in this research and the significance.
2. Missing statistical analysis in the method section.
3. The authors should include a drawing to show the locations of injection and sonication. Figure 2b in this reference they cited can be an example:
https://www.ncbi.nlm.nih.gov/pmc/articles/PMC2974134/
4. The authors may try to snap-freeze the brain without PFA perfusion to check the role of perfusion in the distribution of RBC.
5. To exclude the influence of PFA perfusion on the tracer distribution, the authors can try to extract the brain without perfusion (even without cervical dislocation as it will influence the flow, too) and then post-fix the brain in PFA before sectioning. An example is shown in this paper: https://www.science.org/doi/10.1126/sciadv.aav5447
6. Important control experiments are missing:
(1) use sonication alone without intracortical injection to check how the FUS influences hemorrhage under the current condition.
(2) Also, it will be good to check the BBB integrity after sonication by intravenously injecting fluorescent tracers.
(3) The claim of increased tracer clearance needs further experimental or reference support.
7. The bottom of Figure 3a was cut off.
8. The authors may provide suggestions on avoiding this damage.
Round 2
Reviewer 2 Report (New Reviewer)
The authors addressed my questions well. I don't have more comments.
This manuscript is a resubmission of an earlier submission. The following is a list of the peer review reports and author responses from that submission.
Round 1
Reviewer 1 Report
A very interesting study. Very well designed and presented.
One minor remark: It would be a lot easier for the reader if you presented the details of the experimental groups (strain, gender, age, weight, number of members) at the beginning of Materials and Methods.
"My only remark was that the experimental groups of animals should be more clearly presented at the beginning of Materials and Methods section of the paper."
Reviewer 2 Report
Kim et al. present a study on the presence of brain haemorrhages induced by low intensity FUS application after intracerebral injection.
In the material and method section. Statistical analysis methodology has to be detailed. In the manuscript lines 190-197, it has been stated that an ANOVA was performed. ANOVA is a parametric test, please use a non-parametric test or prove your values follow a normal distribution. I suggest to use a kruskal-wallis test. Lines 198-205 is has been stated that a parametric one-tailed t test was used, could you explain why this test was used and why one-tailed?
The acoustic pressure map show that the focus size is 15mm deep and 3.5mm wide, this focus spot is very big. Could you explain why did you use 200kHz ultrasound ? The authors do not discuss the possibility to create a standing wave in the skull at this frequency. Haemorrhages could be the consequences of inertial cavitation process. A measure of the passive cavitation during FUS should be considered.
The diffusion of the tracers was expected to be increased after using FUS, the volume of fluorescence is lower in the FUS conditions however no diffusion spot is seen which is surprizing. I means all the molecule has diffused to interstitial fluids. This should be checked by measuring the tracers in cerebrospinal fluids.
Reviewer 3 Report
The authors look at the effect of Ultrasound on the brains of post intracranially injected mice. They find that ultrasound increases the spread of the die but also increases damage to the surrounding system. While noting that damage increases with ultrasound, this is not a new idea. As far back as 1999 there were papers discussing the "physical delivery" nature of ultrasound and its potential damaging effects. Numerous reviews also discuss the idea of optimizing US variables to minimize damage. Moreover, US in now being used much more often for the delivery of microbubbles or nanomedicines which also change the context of damage as they US causes rupturing nown to damage surround tissue. Unfortunately for these reasons I do not believe this article has enough novel data to be published in pharmaceutics.
Reviewer 4 Report
The paper assesses the potential of low-intensity transcranial focused ultrasound for delivery of high molecular weight compounds and safety concerns in patients groups with elevated risks associated with weakened or damaged vascular integrity.
1. Please include expected mechanism of permeation in the introduction, include safety concerns/studies already available in literature and the expected mechanism of toxicity.
2. How similar are the healing mechanisms in rat brain to human brain ?
3. How does an injection with a 30g needle in the rat brain compare to CED or other human injection techniques? Is typically ultrasound used immediately after catheter insertion?
4. What would be the difference if catheter remained in the brain and then FUS was used after healing was allowed to take place?
5. Low-intensity FUS was transcranially applied to the injection site shortly after needle 89 retraction to assess the effects of sonication - please define shortly.
6. Please state range of ultrasound used in humans and how its power affects permeability in introduction.
7. Please show power calculation ? Why 15 animals were needed?
8. Please state dose of lidocaine SC injected for local anaesthesia
9. How do you know that 6 minutes are enough to allow the tissue to close? Please reference
10. Why the levels of the fluorescent labelled molecules were not also quantified in slices post imaging to get the total amount of compound?
11. Was haemorrhaging observed only to animals that received sonication? What about haemorrhaging in the control group? Please provide all images as difficult to assess in supplementary information.
12. Is there any literature on how sonication affects clotting or healing? Please elaborate in discussion.
13. Therefore, it is reasonable to postulate that FUS, applied following a 247 sufficient time period after the given local injury may not cause hemorrhaging. I agree with this point, however, you provide no data which would be interesting to the reader to identify how long can sonication be applied after a catheter is inserted.